# Nonlinear and Commutative Editing
# in Pretrained GAN Latent Space

**Takehiro Aoshima**                                    AOSHIMA@HOPF.SYS.ES.OSAKA-U.AC.JP

**Takashi Matsubara**                                    MATSUBARA@SYS.ES.OSAKA-U.AC.JP

*Graduate School of Engineering Science*
*Osaka University*
*1-3 Machikaneyama, Toyonaka, Osaka, 560-8531 Japan*

**Editors:** Sophia Sanborn, Christian Shewmake, Simone Azeglio, Arianna Di Bernardo, Nina Miolane

## Abstract

Semantic editing of images is a fundamental goal of computer vision. While generative adversarial networks (GANs) are gaining attention for their ability to produce high-quality images, they do not provide an inherent way to edit images semantically. Recent studies have investigated how to manipulate the latent variable to determine the images to be generated. However, methods that assume linear semantic arithmetic have limitations in the quality of image editing. Also, methods that discover nonlinear semantic pathways provide editing that is non-commutative, in other words, inconsistent when applied in different orders. This paper proposes a method for discovering semantic commutative vector fields. We theoretically demonstrate that thanks to commutativity, multiple editing along the vector fields depend only on the quantities of editing, not on the order of the editing. We also experimentally demonstrated that the nonlinear and commutative nature of editing provides higher quality editing than previous methods.

**Keywords:** Semantic image editing, GAN, Curvilinear coordinates, Commutativity

## 1. Introduction

The generation and editing of realistic images are one of the fundamental goals in the field of computer vision. Generative adversarial networks (GANs) (Goodfellow et al., 2014) have emerged as a major image generation approach because of the quality of their generated images (Karras et al., 2019, 2020, 2021). However, GANs do not inherently have a way of semantic image editing. Several studies aimed to discover a vector corresponding to an attribute of images and to edit images by adding the attribute vector to the latent variables (Voynov and Babenko, 2020; Härkönen et al., 2020; Shen and Zhou, 2021). These studies introduce the strong assumption of a linear semantic arithmetic on the latent space (see the first column of Table 1), limiting in the quality of image editing. Other studies have proposed to find an attribute vector field in the latent space (Tzelepis et al., 2021; Choi et al., 2022; Ramesh et al., 2018). These methods edit images by integrating a latent variable along the vector field. This approach seems elegant, but edits of different attributes are non-commutative in general. That is, what we get is different when we edit one attribute (denoted by $e_1$) and then edit another (denoted by $e_2$) or when we edit in the reverse order (see the second column of Table 1). This property becomes problematic when one wants to edit various attributes of the same image repeatedly. In contrast, linear arithmetic on the latent space ensures that edits of different attributes are commutative.

To overcome this dilemma, we propose *CurvilinearGANSpace*, which discovers a set of commutative and nonlinear attribute vector fields in pretrained GANs' latent spaces.

Table 1: Comparison of Our Proposal against Related Methods.

| | Linear arithmetic | Vector fields | Proposed |
|---|---|---|---|
| Global coordinate | oblique | (only local) | curvilinear |
| Nonlinear edit | ✗ | ✓ | ✓ |
| Commutative edit | ✓ | ✗ | ✓ |
| Conceptual diagram |  |  |  |

## 2. Methods

### 2.1. Background

Let $\mathcal{X}$ and $\mathcal{Z}$ denote an image space and a GAN latent space, respectively. The generator $G$ of GANs is a mapping from the latent space $\mathcal{Z}$ to the image space $\mathcal{X}$; given a latent variable $z \in \mathcal{Z}$, the generator produces an image $x \in \mathcal{X}$ as $x = G(z)$. We assume the latent space $\mathcal{Z}$ to be an $N$-dimensional space. Let $\{z^i\}_{i=1}^N$ denote the coordinate system (i.e., the basis) on a neighborhood of the point $z \in \mathcal{Z}$. Let $\mathfrak{X}$ denote the set of all vector fields on the latent space $\mathcal{Z}$. Let $X_k \in \mathfrak{X}$ denote a vector field on the latent space $\mathcal{Z}$ indexed by $k$, that is, $X_k : \mathcal{Z} \to \mathcal{T}_z\mathcal{Z}$, where $\mathcal{T}_z\mathcal{Z}$ is the tangent space of the latent space $\mathcal{Z}$ at the point $z$. Then, at the point $z$, the coordinate system on tangent space $\mathcal{T}_z\mathcal{Z}$ is denoted by $\{\frac{\partial}{\partial z^i}\}_{i=1}^N$, and a vector field $X_k$ is expressed as $X_k(z) = \sum_{i=1}^N X_k^i(z)\frac{\partial}{\partial z^i}$ for smooth functions $X_k^i : \mathcal{Z} \to \mathbb{R}$.

When considering a method that assumes attribute vector fields (e.g., Tzelepis et al. (2021); Choi et al. (2022); Ramesh et al. (2018)), an edit of an attribute $k$ of an image $x$ is done by integrating a latent variable $z$ along the corresponding vector field $X_k$; the edited image is given by $x' = G(z')$ for $z' = z + \int_0^t X_k(z(\tau))\mathrm{d}\tau = \phi_k^t(z)$, where $\phi_k^t$ denotes the flow that arises from the vector field $X_k$. Edits of two attributes $k$ and $l$ are commutative if and only if two flows are commutative, that is, $\phi_l^s \circ \phi_k^t = \phi_k^t \circ \phi_l^s$ for any $s, t \in \mathbb{R}$ at any point $z \in \mathcal{Z}$. In general, two vector fields are non-commutative, and hence two edits are non-commutative (Lee, 2012). A method that assumes linear attribute arithmetic (e.g., Voynov and Babenko (2020); Härkönen et al. (2020); Shen and Zhou (2021)) can be regarded as a special case. Using an attribute vector $a_k$ independent of the position $z$, a vector field can be defined as $X_k(z) \equiv a_k$, and then its flow is $\phi_k^t(z) = \int_0^t a_k\mathrm{d}\tau = t\,a_k$. Edits are commutative, but the quality of image editing is limited due to the linearity.

### 2.2. CurvilinearGANSpace

We introduce the following theorem of differential geometry (see Lee (2012) for example).

**Theorem 1** *Let vector fields $X_1, X_2, \ldots, X_N$ on an $N$-dimensional space $\mathcal{Z}$ be linearly independent and commutative on an open set $\mathcal{U} \subset \mathcal{Z}$. At each $z \in \mathcal{U}$, there exists a smooth coordinate chart $\{\frac{\partial}{\partial s^i}\}_{i=1}^N$ centered at $z$ such that $\frac{\partial}{\partial s^i} = X_i$.*

Roughly speaking, a set of linearly independent and commutative vector fields is compatible with a set of vector fields along the axes of a coordinate system up to geometric transfor-

mation. Hence, we consider the case where the open set $\mathcal{U}$ in Theorem 1 is not a proper subset but equal to the latent space $\mathcal{Z}$.

We prepare an $N$-dimensional Euclidean space $\mathcal{V}$ and name it the Cartesianized latent space. Its coordinate system $\{v^i\}_{i=1}^N$ is a global Cartesian coordinate system. Let $e_k$ denote the $k$-th element of the standard basis, and the vector filed $\tilde{X}_k$ corresponding to the attribute $k$ is defined as $\tilde{X}_k := e_k$ for $e_k := \frac{\partial}{\partial v^k}$. The flow $\psi_k : \mathbb{R} \times \mathcal{V} \to \mathcal{V}$ the arises from the vector filed $\tilde{X}_k$ is given by $\psi_k^t(v) := v + \int_0^\tau e_k \mathrm{d}\tau = v + t\,e_k$. Obviously, the flows $\psi_k$ are commutative because $\psi_l^s \circ \psi_k^t(v) = v + t\,e_k + s\,e_l = \psi_k^t \circ \psi_l^s(v)$. We introduce a smooth bijective mapping $f : \mathcal{Z} \to \mathcal{V}, z \mapsto v$, corresponding to the coordinate chart in Theorem 1. We define a flow $\phi_k^t$ that edits the attribute $k$ on the latent space $\mathcal{Z}$ as $\phi_k^t := f^{-1} \circ \psi_k^t \circ f$. A vector field $X_k$ on the latent space $\mathcal{Z}$ is implicitly defined by pushforwarding the vector field $\tilde{X}_k$ on the Cartesianized latent space $\mathcal{V}$; in particular, $X_k(z) = (f^{-1})_*(\tilde{X}_k) = \frac{\partial f^{-1}(v)}{\partial v} e_k$ at the point $z$ for $v = f(z)$. Then, one can generate an edited image $x' = G(z')$ using the generator $G$. A coordinate system defined by a bijective transformation of a Cartesian coordinate is called a curvilinear coordinate (Arfken et al., 2012). Hence, we name this method CurvilinearGANSpace.

CurvilinearGANSpace is a commutative special case of method that assumes attribute vector fields (e.g., Choi et al. (2022); Ramesh et al. (2018); Tzelepis et al. (2021)). At the same time, it is a nonlinear generalization of method that assumes attribute arithmetic (e.g., Voynov and Babenko (2020); Härkönen et al. (2020); Shen and Zhou (2021)); CurvilinearGANSpace enjoys the advantages of both methods; the nonlinearity and commutativity.

## 3. Experiments and Results

**Experimental Settings** The proposed methodology is available for any framework that manipulates the latent variables. This paper focuses on the unsupervised learning framework proposed by Voynov and Babenko (2020). We used CelebA-HQ (Liu et al., 2015) as the dataset, StyleGAN2 (Karras et al., 2020) for the GANs, and ResNet-18 (He et al., 2016) for the reconstructor used in the learning framework. For a smooth bijective mapping $f$, we employ a continuous normalizing flow (CNF) (Chen et al., 2018). For comparison, we also evaluated a method that assumes a linear arithmetic (Voynov and Babenko, 2020) and a method that assumes vector fields called WarpedGANSpace (Tzelepis et al., 2021). To clarify the difference, we will refer to the former method as LinearGANSpace, hereafter. We used their pretrained models.

**Evaluation Metrics** Even when editing an attribute $k$ of a latent variable $z$ by a change amount $t$, it is not guaranteed that the same attribute $k$ of the image $x$ is edited by the same amount $t$. Hence, we normalized the change amount $t$ for the latent variable $z$ by the change amount for the generated image $x$, following the measurements by a separate attribute predictor $A_k(\cdot)$. We used CelebA-HQ attributes classifier for smiling and bangs (Jiang et al., 2021) and Hopenet for face direction (yaw) (Doosti et al., 2020). As well as attributes, we used ArcFace for the identity score $I(\cdot, \cdot)$, evaluating whether two images are of the same person (Deng et al., 2019). We defined *commutativity error* of attributes $k + l$ to evaluate how commutative image editing. It is the error when edits of these two attributes $k$ and $l$ are applied in different orders; namely, $|A_k(G(\phi_k^t(\phi_l^t(z)))) - A_k(G(\phi_l^t(\phi_k^t(z))))|$ and

Table 2: Results of StyleGAN2. S: Smiling, B: Bangs, Y: Yaw.

| | Commutativity Error [%] | | | | | | Identity Error [%] | | |
|---|---|---|---|---|---|---|---|---|---|
| | S+B | | S+Y | | B+Y | | S | B | Y |
| LinearGANSpace | **0.05** | **0.04** | **0.04** | **0.02** | **0.03** | **0.06** | 14.36 | 14.70 | 17.64 |
| WarpedGANSpace | 21.67 | 24.47 | 13.47 | 2.79 | 19.94 | 4.68 | 5.29 | 22.05 | **7.01** |
| CurvilinearGANSpace (proposed) | 0.25 | 0.17 | 0.27 | 0.36 | 0.20 | 0.31 | **4.98** | **9.29** | 10.19 |

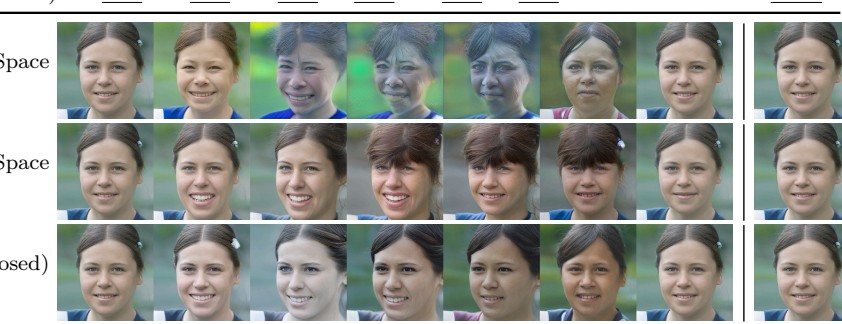

Original  +Smiling  +Yaw  +Bangs  −Smiling  −Yaw  −Bangs        Original
Figure 1: Visualization results.

$|A_l(G(\phi_k^t(\phi_l^t(z)))) - A_l(G(\phi_l^t(\phi_k^t(z))))|$. We also defined *identity error* of attribute $k$ to evaluate the editing quality. It measures how much an edit of the attribute $k$ reduces the identity score; namely, $1 - I(G(z), G(\phi_k^t(z)))$. We set the change amount of every edit to $t = 0.1$ for the smiling and bangs attribute and $t = 5$ degrees for the yaw attribute. We divided the commutativity errors by the change amount $t$ and showed them in percentages.

**Results**   We summarized the numerical results in Table 2. WarpedGANSpace produced the commutativity errors of at least 2.79 % and often more than 10 %. Those of LinearGANSpace and the proposed CurvilinearGANSpace were always less than 0.5 %; while not exactly zero due to numerical and rounding errors, they are negligible. Hence, as expected, edits by WarpedGANSpace are not commutative, and edits by LinearGANSpace and CurvilinearGANSpace are commutative. CurvilinearGANSpace produced the lowest identity errors for smiling and bangs attributes and the second lowest one for the yaw attribute. Hence, we can say that it learned disentangled representations better.

We showed an example image with a sequence of edits in Fig. 1. The amount of change was set to double to make the change easier to find; +Smiling indicates that $t = +0.2$ and $k =$ Smiling. LinearGANSpace's image editing quality is clearly inferior. The image editing qualities of WarpedGANSpace and CurvilinearGANSpace are competitive. After six edits, the change amounts should cancel out, and the attributes should return to their original values. LinearGANSpace and CurvilinearGANSpace show the expected results. However, the edited result of WarpedGANSpace shows that the woman's mouth opening has not returned to its original state; edits by WarpedGANSpace are not commutative. Therefore, we conclude that CurvilinearGANSpace enjoys the advantages of both of previous methods; the nonlinearity and commutativity.

## Acknowledgments

This study was partially supported by JST PRESTO (JPMJPR21C7), JST CREST (JP-MJCR1914), and JSPS KAKENHI (19H04172, 19K20344), Japan.

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
