# OpenReview forum: "Nonlinear and Commutative Editing in Pretrained GAN Latent Space"
_NeurIPS.cc/2022/Workshop/NeurReps — NeurReps 2022 Poster_

### Official Review · Reviewer_fLmp · 2022-10-07
**A very clear, solid paper with solid technical results that I enjoyed reading**

**Confidence:** 4
**Soundness:** 4
**Presentation:** 4
**Contribution:** 3
**Overall Rating:** 7

**Summary:**

The paper builds upon previous linear and non-commutative frameworks for semantic editing in latent space, which allows them to build a framework which is both nonlinear and commutative with respect to these semantic edits.

**Questions:**

-Sections 2.1 and 2.2 are very dense and a bit difficult to read, even visually. While I understand the constraints on the space of the paper, I think this could benefit from reducing these sections to high-level mathematical results that make the model differences explicit so that the reader doesn't have to parse the equations for what about your derivations are different, and then adding more details in the appendix.

-The originality of theorem 1, a main theorem of the text as I understand it, is unclear to me by the wording "we introduce..." This appears to be taken from a textbook? If so, I would change the wording of this.

-Are these methods too computationally expensive to run multiple trials to get error bars for Table 2? How stable are these results?

-As a non-practitioner in this field, I still don't quite understand why we care about non-commutativity if the identity error is sufficiently low. Isn't editing quality the main evaluation metric for semantic editing? WarpedGan is fairly competitive with your method if so. I understand the mathematical desirability, but the practical considerations are less clear to me

**Limitations:**

Stability of results are, in my opinion, the main limitation. Are the percentages means over many trials? How are these obtained? Even just a few sentences on training details could be helpful.

**Recommended Decision:**

3: Accept

**Relevance:**

4: Highly relevant

**Strengths And Weaknesses:**

-Table 1 is very helpful for the reader to understand what exactly the paper is attempting to do. The paper solves a clearly defined problem. It is well written and set up professionally.

-See more specific comments below about originality. It seems to me that you are using previous theorems but in a novel way. This could use some clarification. In any case, the results are indeed promising and introduces original properties of a computational model.

-See more specific comments below about stability of main results presented in Table 2. The paper is otherwise very technically sound.

-I otherwise have no glaring weaknesses that stand out to me. Great job overall!


**Submission Track:**

Extended Abstract (4 Page)

---

### Official Review · Reviewer_PSYg · 2022-10-13
**Curvilinear Editing in the GAN Latent Space**

**Confidence:** 4
**Soundness:** 3
**Presentation:** 4
**Contribution:** 2
**Overall Rating:** 5

**Summary:**

This paper aims to solve the problem of semantic editing in GANs. GANs are famous for their reconstruction fidelity, however, GAN latent space is not very easy to edit. If we assume linear semantic arithmetic on the latent space, often the quality of the images generated after the editing is not that great. However, operations on the latent space are then commutative which means that they can be reversed and changing the order of operations still give us the same results. Similarly, to edit the latent space, we can find an attribute vector field in the latent space and edit images by integrating a latent variable along this vector field. Although it yields images of higher quality it is not commutative.

To solve this problem authors propose CurvilinearGANSpace, which allows for nonlinear edits while keeping the operations commutative in nature. The proposed method finds the attribute vector fields in the GAN latent spaces that are both commutative and nonlinear.

**Questions:**

- I would like to know more about the limitations of this method, where it fails and where it works. If I can know more about what can be done with this (or what do you plan to do with this), I am willing to revaluate my scores.

**Limitations:**

The authors do not address the limitations of their work adequately.

**Recommended Decision:**

1: Reject

**Relevance:**

3: Solid fit

**Strengths And Weaknesses:**

**Strengths** : The paper written is of high quality as the mathematics behind the method is explained pretty well. The experiments involves comparison with other competing methods and include both qualitative and quantitative results which makes the author's claim stronger. Overall the claims are well supported in mathematical foundations as well as empirical results.

**Weaknesses** : I think the main contribution of this paper is not very impactful to current research landscape. I believe there are some stochastic differential equation editors for diffusion models (which is different than GANs I suppose). Nonetheless it is an interesting innovation to GANs.

**Submission Track:**

Extended Abstract (4 Page)

---

### Official Review · Reviewer_W5Ec · 2022-10-13
**Interesting work on latent space arithmetic**

**Confidence:** 4
**Soundness:** 3
**Presentation:** 4
**Contribution:** 3
**Overall Rating:** 8

**Summary:**

The paper is concerned with the question of how to "edit" images represented as latent variables in a GAN. While the work is using GANs, the work should extend to any choice of deterministic generator. The key issue addressed by the work is that ordinary linear edits are sensitive to the order of the edits. The paper proposes to learn vector fields in the latent space and push edits along these. This is a neat solution.

**Questions:**

I wonder, can the idea of learning vector fields be viewed as learning a metric on the latent space? That would render the work more general and applicable to more tasks. Others have learned metrics over latent spaces (see e.g. "Latent Space Oddity", Arvanitidis et al., ICLR 2018), but these metrics appear to be task-independent. In contrast, the proposed work is tailored for a given task (potentially making it more useful).

**Limitations:**

The main issue with the presented work is, in my view, that the experiments are quite limited. This is, however, perfectly fine for a submission of this type, and I encourage the authors to continue the work.

**Recommended Decision:**

3: Accept

**Relevance:**

3: Solid fit

**Strengths And Weaknesses:**

Strengths:
* The paper is well-written and easy to follow.
* The work addresses questions that are of general interest in the CV/ML community.
* The idea appears both sensible and novel.
* The geometric nature of the work makes it a good fit for the workshop.

Weaknesses:
* The paper talks exclusively about GANs, but I think the work is more general. I recommend the authors clarify this in the paper.
* Experiments are limited and at a proof-of-concept stage, but I think this is exactly the point of writing an extended abstract (to get early feedback).


**Submission Track:**

Extended Abstract (4 Page)

---

### Decision · Program_Chairs · 2022-10-21

Accept (Poster)